# DFT Study of the Direct Radical Scavenging Potency of Two Natural Catecholic Compounds

**DOI:** 10.3390/ijms232214497

**Published:** 2022-11-21

**Authors:** Ana Amić, Denisa Mastiľák Cagardová

**Affiliations:** 1Department of Chemistry, Josip Juraj Strossmayer University of Osijek, Ulica cara Hadrijana 8A, 31000 Osijek, Croatia; 2Institute of Physical Chemistry and Chemical Physics, Department of Chemical Physics, Slovak University of Technology in Bratislava, Radlinského 9, SK-812 37 Bratislava, Slovakia

**Keywords:** quercetin, rooperol, peroxyl radicals, density functional theory, kinetics, tunnelling

## Abstract

To ascertain quercetin’s and rooperol’s potency of H-atom donation to CH_3_OO^•^ and HOO^•^, thermodynamics, kinetics and tunnelling, three forms of chemical reaction control, were theoretically examined. In lipid media, H-atom donation from quercetin’s catecholic OH groups via the proton-coupled electron transfer (PCET) mechanism, is more relevant than from C-ring enolic moiety. Amongst rooperol’s two catecholic moieties, H-atom donation from A-ring OH groups is favored. Allylic hydrogens of rooperol are poorly abstractable via the hydrogen atom transfer (HAT) mechanism. Kinetic analysis including tunnelling enables a more reliable prediction of the H-atom donation potency of quercetin and rooperol, avoiding the pitfalls of a solely thermodynamic approach. Obtained results contradict the increasing number of misleading statements about the high impact of C–H bond breaking on polyphenols’ antioxidant potency. In an aqueous environment at pH = 7.4, the 3-O^−^ phenoxide anion of quercetin and rooperol’s 4′-O^−^ phenoxide anion are preferred sites for CH_3_OO^•^ and HOO^•^ inactivation via the single electron transfer (SET) mechanism.

## 1. Introduction

Epidemiological evidence and traditional knowledge suggest that diets rich in polyphenolic compounds, such as the Mediterranean diet (characterized by regular intake of fruit, vegetables, fish, nuts, cereals, olive oil and red wine) are associated with good health and reduced risk for numerous chronic diseases. The etiology of mutagenesis, carcinogenesis and cardiovascular and neurodegenerative disorders is related to the cellular damage caused by the overproduction of radicals in oxidative stress conditions, when endogenous enzymatic defense mechanisms are not able to combat excess radicals [1]. In this case, exogenous antioxidants, such as nutritional polyphenols, may help to restore homeostasis. Among various possible mechanisms of the protective action of polyphenols, direct scavenging of radicals was indicated as operative [2]. In order to be an in vivo active direct radical scavenger, a polyphenolic compound must be bioavailable and reach a sufficiently high concentration in systemic circulation. Traditionally, the phenolic -OH group was assumed as the main structural feature related to effective radical scavenging by polyphenols [3]. Catechol moiety (*ortho*-dihydroxy group) in the B-ring and/or the enolic 3-OH group offer the highest antiradical potency to flavonoids, a subgroup of polyphenols. In other natural compounds without phenolic O–H groups, such as polyunsaturated fatty acids (PUFAs, e.g., linoleic acid), the allylic C–H moiety is active in radical inactivation [4].

In this report, we considered the thermodynamic and kinetic aspects of peroxyl radical scavenging by two natural polyphenolic compounds: quercetin and rooperol (Figure 1). Quercetin, one of the most studied flavonoids, appears ubiquitously as a glycoside in dietary fruits and vegetables, especially in yellow and red onions. Despite its generally poor bioavailability, a plethora of quercetin’s benefits on human health was suggested including cardioprotective, anti-inflammatory and anticancer activities [5]. It is also well-known as a supreme in vitro radical scavenger. Rooperol ((*E*)-1,5-*bis*(3′,4′-dihydroxyphenyl)pent-1-en-4-yne) is the aglycone of hypoxoside, a major bioactive compound derived from the *Hypoxis rooperi* plant (African potato) traditionally used in folk medicine [6]. Rooperol possesses various biological activities, such as antitumor and antibacterial, and reveals antioxidant potential via metal chelation and radical scavenging [7].

The major structural features of quercetin as an antioxidant are catechol and 3-OH enolic moiety, while it is expected that rooperol’s two catechol groups have radical scavenging potency superior to its allylic hydrogens (Figure 1). We investigated the contribution of those structural features to antiradical potency in non-polar and polar media using pentyl ethanoate and water as solvents to mimic lipid (cell membrane) and aqueous environments, respectively. The mechanism and kinetics of quercetin reactions with peroxyl radicals in water solution were studied by using both an experimental and theoretical approach [8]. Recently, the thermodynamics of rooperol’s antioxidant potency in polar and non-polar media was reported [9]. In non-polar media, formal hydrogen atom transfer (fHAT) is the dominant mechanism in radical inactivation. Depending on the presumed active bond of an antioxidant (O–H or C–H), fHAT may proceed as proton-coupled electron transfer (PCET) or ‘pure’ HAT, respectively. In the PCET mechanism, the proton and electron are transferred in a single elementary step between different donor and acceptor sites, whereas in the HAT mechanism, the electron and proton are transferred together as a hydrogen atom between the same donor/acceptor sites [2,10].

The rate of an fHAT reaction of a polyphenol (PhOH) with peroxyl (^•^OOR) depends on the barrier height for hydrogen atom transfer from PhOH to ^•^OOR, shown in Equation (1):(1)$$\mathrm{PhOH}\;+{\;}^{•}\mathrm{OOR}\overset{k}{\to }{\mathrm{PhO}}^{•}\;+\;\mathrm{HOOR}$$

As the reaction becomes more exergonic (more negative reaction Gibbs free energy, Δ_r_*G*), the barrier height (activation Gibbs free energy, Δ*G*^≠^) should decrease, and the polyphenol (antioxidant) reacts faster with ^•^OOR, thus preventing reaction with the substrate. Consequently, it is clear that the rate constant *k* for Equation (1) is the key factor in evaluating the potency of polyphenols as radical scavengers. The experimental determination of antioxidant kinetics is a somewhat delicate task to be systematically performed for many polyphenols and radicals under different conditions. Quantum chemical kinetic calculations based on density functional theory (DFT) are a relevant alternative for comprehensive evaluation of radical scavenging potency [11].

Less demanding is the computational thermodynamic approach, which implies that the activation energy for the H-atom transfer is proportional to the strength of the O–H bond. Thus, the bond dissociation enthalpy (BDE) appears to be an important physical parameter in determining the antioxidant potency of polyphenols, since the weaker the O–H bond, the faster the reaction with radicals [12]. BDE is defined by Equation (2):BDE = *H*(PhO^•^) + *H*(H) − *H*(PhOH)(2)
where *H*(PhO^•^), *H*(H) and *H*(PhOH) are enthalpies of the phenoxyl radical, H-atom and parent phenolic antioxidant, respectively. Two decades ago, BDE was recognized as an excellent primary indicator of antioxidant activity [12]. A lower value of BDE is associated with effective radical scavenging. Theoretically calculated BDE values using appropriate theoretical levels correlate well with experimentally assayed ones for simple phenolics [13]. Because of the lack of experimental BDE values for a vast majority of polyphenols, calculated BDE values have been used in the estimation of their antioxidant potency [2,14,15]. Earlier, such calculations were mainly performed using the B3LYP functional, which has since been recognized as inappropriate because it systematically underestimates BDE values. Hybrid Minnesota density functionals, such as M05-2X and M06-2X, represent a good choice regarding computational cost and accuracy [16].

In accordance with the Evans–Polanyi principle, the logarithms of the *k* constants are inversely related to the BDEs of polyphenols but not fully correlated [2]. Only if the same radical and the same family of phenols are considered in the same solvent, O–H BDE is fully correlated with the rate constant [14]. Similarly, the C–H BDE predicts the reactivity for HAT reactions correctly involving only one type of oxidant X^•^ reacting with the same family of compounds [17]. Hence, predictions of the antioxidant potency based solely on thermodynamic descriptors, such as BDE, must be taken with caution. However, the literature is continuously filled with reports dealing solely with the thermodynamics of the radical scavenging potency of polyphenolics.

The main reason for the inadequacy of such predictions is that the decisive role of the barrier heights and the contribution of tunnelling cannot be recognized when just BDE data are handled [18]. Tunnelling is recognized as the third form of chemical reaction control next to thermodynamic and kinetic control [19]. A common misconception is that tunnelling is important only at cryogenic temperatures. It is now well-established that tunnelling contributions may be substantial at room temperature and under physiological conditions [20]. Thus, the final decision of the radical scavenging potency of polyphenolic compounds should be guided by kinetics, including tunnelling contribution [10,16]. To achieve this, the popular B3LYP functional should be neglected because it highly underestimates the barrier heights and is a poor choice for kinetic calculations [16].

In this report, the effectiveness of quercetin and rooperol in the inactivation of methyl peroxyl radical (CH_3_OO^•^) and hydroperoxyl radical (HOO^•^) in lipidic and aqueous media was investigated. CH_3_OO^•^ may serve as the simplest model of damaging lipid peroxyl radicals (LOO^•^), which are abundantly formed in biological systems. HOO^•^ was investigated because it is the simplest of the biologically relevant peroxyl radicals with intermediate reactivity and a not-too-short half-life to be effectively scavenged by polyphenolic compounds [21]. HOO^•^ can initiate peroxidation of fatty acids, resulting in membrane lipid damage that may underlay degenerative diseases and aging.

The main goal of the present work was to analyze the thermodynamics, kinetics and tunnelling underlying H-atom transfer reactions by which quercetin and rooperol may inactivate CH_3_OO^•^ and HOO^•^ radicals in lipidic media. Particular attention was devoted to the relevance of rooperol’s phenolic O–H vs. allylic C–H bond breaking. To that purpose, quantum chemical calculations of relevant thermodynamic and kinetic data, as well as of tunnelling corrections, were performed. Single electron transfer (SET) reactions of phenoxide anions of quercetin and rooperol were investigated in aqueous media. Detailed analysis of obtained results indicated preponderant antiradical moiety and the underlying mechanism.

## 2. Results and Discussion

### 2.1. PCET from Phenolic O–H Groups of Quercetin to CH_3_OO^•^ and HOO^•^ Radicals

Although the presented investigation is based on the kinetic analysis including tunnelling, i.e., theoretical evaluation of the rate constants, relevant thermodynamic data were also calculated and considered. In Table 1, the values of BDE, reaction Gibbs free energy (Δ_r_*G*), TS imaginary frequency (*ν*), barrier heights (Δ*G*^≠^), TST rate constant *k*^TST^, Eckart (Wigner) tunnelling corrections (*κ*^Eck^, *κ*^Wig^) with related rate constants (*k*^TST/Eck^, *k*^TST/Wig^) and branching ratio Γ (%) for the PCET paths of quercetin reactions towards CH_3_OO^•^ and HOO^•^ radicals are presented.

The lowest numerical value of BDE, Δ_r_*G* and Δ*G*^≠^ (and the highest *k*^TST^ value) may indicate the preferred reaction path for H-atom donation from quercetin’s OH groups. By closer inspection of the data presented in Table 1, it is clear that for both CH_3_OO^•^ and HOO^•^ radicals, the preferred reaction path is via catecholic moiety (3′-OH and 4′-OH group) followed by the 3-OH group. It should be noted that numerous reports on radical scavenging by polyphenolic compounds have used solely O–H BDE to ascertain the thermodynamically favored reaction path. Some of them have presumed kinetic feasibility of this path because a high correlation between O–H BDE and log *k* may exist [14,17]. We already noted that such an approach could be questionable. Therefore, we correlated data presented in Table 1 (i.e., BDE vs. Δ_r_*G*, Δ*G*^≠^, *k*^TST^, *k*^TST/Eck^ and *k*^TST/Wig^) and graphically present results in Appendix A. Obtained high correlations (|*r*| > 0.92 in the case of CH_3_OO^•^ and |*r*| > 0.94 in the case of HOO^•^) indicate BDE as a usable descriptor of reactivity. This is a reasonable outcome because BDE of the same type of bonds, i.e., phenolic O–H bonds, was considered [14,17].

Any obtained theoretical prediction must be in line with experimental facts. Recently, experimental ESR measurements indicated that the unpaired electron of quercetin radical is mostly delocalized in the B-ring and partly on the AC rings [22]. Thus, in the case of CH_3_OO^•^ and HOO^•^ scavenging by quercetin, the BDE correctly predicts catecholic moiety as the preferred site for H-atom donation from both thermodynamic and kinetic points of view. We emphasize this fact because if diverse types of bonds are jointly considered, BDE is not appropriate as a reactivity descriptor, as is discussed in Section 2.2.

Klippenstein et al. emphasized that any H-atom transfer reaction or proton transfer reaction with a Δ*G*^≠^ of several kcal/mol or more is probably dominated by tunnelling at room temperature [23]. A recent experimental study by Nakanishi et al. indicated notable quantum mechanical tunnelling in the hydrogen-transfer reaction from the phenolic O–H group of Trolox to the 2,2-diphenyl-1-picrylhydrazyl (DPPH^•^) radical in aqueous media at ambient temperature [24], in line with the earlier observed fact that tunnelling plays an important role in scavenging of lipid peroxyl radicals by vitamin E [25]. We theoretically estimated the tunnelling contribution to the *k*^TST^ by using two one-dimensional methods (Wigner and Eckart) as implemented in the Eyringpy program [26]. The simplest method is Wigner’s, which only requires imaginary frequency at the TS. The Eckart method represents the barrier shape more accurately and can reproduce the results of multi-dimensional methods [27], but its accuracy noticeably depends on the reaction system. We found that the Eckart tunnelling corrections enhance the $k_{\mathrm{overall}}^{\mathrm{TST}}$ from 1.2 × 10^1^ M^−1^ s^−1^ to $k_{\mathrm{overall}}^{\mathrm{TST}/\mathrm{Eck}}$ = 8.3 × 10^2^ M^−1^ s^−1^ (Table 1), indicating the significance of the tunnelling contribution to H-atom donation from phenolic OH groups to CH_3_OO^•^. Similarly, Wigner tunnelling corrections enhance the $k_{\mathrm{overall}}^{\mathrm{TST}}$ from 1.2 × 10^1^ M^−1^ s^−1^ to $k_{\mathrm{overall}}^{\mathrm{TST}/\mathrm{Wig}}$ = 7.0 × 10^1^ M^−1^ s^−1^. Tunnelling appears much more pronounced for H-atom transfer from the 3-OH group, but due to the lowest barrier heights for catecholic 3′-OH and 4′-OH reaction paths, the rate constants for the later paths dominate the overall rate.

The importance of each individual reaction path can be easily predicted by using branching ratios (the relative amount of products, Γ (%)). Both Eckart (Γ^Eck^) and Wigner (Γ^Wig^) tunnelling approaches indicate catecholic moiety of quercetin (Γ^Eck^ = 56.4% + 43.2% = 99.6%; Γ^Wig^ = 60.0% + 40.0% = 100%) as contributing most to the inactivation of CH_3_OO^•^.

However, as can be seen from the lower part of Table 1, where the results of HOO^•^ inactivation by quercetin are presented, Γ^Eck^ indicates H-atom donation from the 3-OH group to HOO^•^ as the preferable reaction path (Γ^Eck^ = 83.0%), while Γ^Wig^ retains catecholic moiety as the preferable scavenging moiety (Γ^Wig^ = 52.9% + 40.4% = 93.3%). A reasonable explanation of this prediction could be that the Eckart method overestimates tunnelling for the H-atom abstraction from the 3-OH group, supported by the fact that this method tends to overestimate the tunnelling contribution [28]. This feature is also visible in Figure 2a, where the contribution of tunnelling corrections to the kinetics of CH_3_OO^•^ scavenging by quercetin is shown but has negligible influence on the $k_{\mathrm{overall}}^{\mathrm{TST}/\mathrm{Eck}}$ value. In the case of HOO^•^ scavenging (Figure 2b), overestimated tunnelling has a significant impact on the calculated $k_{\mathrm{overall}}^{\mathrm{TST}/\mathrm{Eck}}$ value of 2.9 × 10^3^ M^−1^ s^−1^ (Table 1). It is worth mentioning that a huge *κ*^Eck^ value was obtained by considering RC and PC in evaluation of the Eckart tunnelling contributions for the H-atom abstraction from the 3-OH group (Appendix A).

Thus, the calculated $k_{\mathrm{overall}}^{\mathrm{TST}/\mathrm{Eck}}$ value of 2.9 × 10^3^ M^−1^ s^−1^ should be taken with caution. By using data presented in Table 1 and shown in Figure 2b, a linearly approximated *k*^TST/Eck^ value for the C3-OH site amounts to 1.2 × 10^1^ M^−1^ s^−1^, which (instead of 2.4 × 10^3^ M^−1^ s^−1^) contributes to the more reliable predicted $k_{\mathrm{overall}}^{\mathrm{TST}/\mathrm{Eck}}$ value of 5.0 × 10^2^ M^−1^ s^−1^.

To the best of our knowledge, there are no published experimental results of CH_3_OO^•^ and HOO^•^ inactivation by H-atom donation from quercetin via the PCET mechanism. Due to this, we compared our theoretically predicted results with assayed ones obtained using another radical. Our predicted results for scavenging of CH_3_OO^•^ ($k_{\mathrm{overall}}^{\mathrm{TST}/\mathrm{Eck}}$ = 8.3 × 10^2^ M^−1^ s^−1^) and HOO^•^ ($k_{\mathrm{overall}}^{\mathrm{TST}/\mathrm{Eck}}$ = 5.0 × 10^2^ M^−1^ s^−1^) are in good agreement with the experimentally assayed results for the reaction of quercetin with DPPH^•^, which amounts to 4.76 × 10^2^ M^−1^ s^−1^, in methanol solution at 25 °C [29] as well as with the published results (1.08 × 10^3^ M^−1^ s^−1^) related to the hydrogen-transfer reaction from quercetin to galvinoxyl radical (an oxyl radical species) [30]. They are also comparable with previously calculated results for CH_3_OO^•^ using a different theoretical approach. By using canonical variational transition state theory (CVT) with small-curvature tunnelling (SCT) computed at the MPWB1K/6-311G** level of theory at 300 K in gas phase, the obtained rate constant amounts to *k*^CVT/SCT^ = 9.63 × 10^3^ M^−1^ s^−1^ [31]. Regarding scavenging of HOO^•^, our results are consistent with the published results using the M05-2X/6-311+G(d,p) level of theory in gas phase, indicating catechol moiety of quercetin as the most active site in H-atom donation [32]. Thus, despite observed shortcomings, Eckart tunnelling corrections arise as more acceptable than Wigner’s because they better match the abovementioned experimental results.

Data presented in Table 1 include reaction paths via all phenolic OH groups. For both HOO^•^ and CH_3_OO^•^ radicals, the contribution of 5-OH and 7-OH reaction paths to the scavenging potency of quercetin is negligible. Therefore, we did not further consider these reaction paths. Despite this, data given in Table 1 refer to the results of laborious work for all phenolic OH groups.

### 2.2. PCET from Phenolic O–H and HAT from Allylic C–H Groups of Rooperol to HOO^•^ and CH_3_OO^•^ Radicals

Rooperol has three moieties with possible radical scavenging activity (Figure 1): two catecholic groups, which are traditionally known as the supreme H-atom (electron) donation site, and allylic moiety, for which recent claims state that it may exert a considerable contribution to polyphenols’ radical scavenging potency. The validity of such claims can be evaluated from Table 2a, where calculated values of BDE, Δ_r_*G*, *ν*, Δ*G*^≠^, *k*^TST^, *κ*^Eck^, *k*^TST/Eck^, *κ*^Wig^, *k*^TST/Wig^ and Γ for the phenolic and allylic H-atom donation paths from rooperol to HOO^•^ radical are presented.

Taking together phenolic and allylic reaction paths, thermodynamic parameters BDE and Δ_r_*G*, which are highly correlated (*r* > 0.99, Appendix A), both indicate allylic hydrogens as the most abstractable. On the contrary, kinetic parameters Δ*G*^≠^ and *k*^TST^ (whose correlations with clusters of O–H and C–H BDE values (i.e., BDE of different types of bonds) show reasonless trends (Appendix A)) indicate catecholic hydrogens as the most abstractable by HOO^•^. Therefore, thermodynamics and kinetics herein give opposed predictions. Regarding experimental facts, it is well-known that main-group radicals (such as peroxyl radicals, ROO^•^) abstract H-atom much faster (~10^4^) from the phenolic O–H bond than from the C–H bond of similar BDE value [10,17]. This clearly illustrates that the BDE is not the sole or final criterion to ascertain which structural motif of an antioxidant molecule contributes the most to radical scavenging [10,19].

As noted in the Introduction section, the barrier heights and quantum mechanical tunnelling effects are those responsible for making the reaction between an antioxidant and radical fast [17,18]. We emphasized this fact because of very recently published misleading predictions of higher aliphatic (allylic, benzylic) C–H vs. phenolic O–H reactivity based on thermodynamic analysis, i.e., consideration of BDE values (for example, see [9]). Such proposals illustrate the pitfalls of estimating rates of H-atom transfer reactions by analyzing only thermodynamics. Kinetic analysis should be performed because it gives a more realistic picture of the radical scavenging potency of polyphenolic compounds [10,11,16].

The total rate constant for catecholic moieties of rooperol ($k_{\mathrm{total}}^{\mathrm{TST}/\mathrm{Eck}}$) is equal to the sum of the rate constants (*k*^TST/Eck^) of all OH reaction paths (3′-OH, 4′-OH, 3″-OH and 4″-OH). It amounts to 1.4 × 10^3^ M^−1^ s^−1^. For two allylic hydrogens, this sum amounts to 3.0 × 10^0^ M^−1^ s^−1^ (Table 2a). Those values indicate that the H-atom abstraction from catecholic moieties is ~10^3^ times faster than from allylic hydrogens, in good accordance with known facts about phenolic O–H vs. C–H bond reactivity [10,17]. Analogous results regarding O–H vs. C–H reactivity have been recently estimated for equol [33], dihydrokaempferol [34], mactanamide and lariciresinol [35]: phenolic hydrogens contribute the most for HOO^•^ capture and are 10^2^ to 10^3^ times more abstractable than aliphatic (benzylic, allylic) hydrogens. According to the estimated branching ratios, only products yielded by phenolic reaction paths are expected to be formed (Γ^Eck^ = 99.8%), Table 2a. The involvement of the allylic reaction paths is negligible (Γ^Eck^ = 0.2%).

An interesting feature is that rooperol’s two catecholic moieties are not equally efficient in HOO^•^ and CH_3_OO^•^ inactivation. Namely, the contributions (Γ^Eck^) of the phenolic OH groups in the A and B-ring of rooperol are 89.6% and 10.2%, respectively, for HOO^•^ scavenging (Table 2a), and 81.4% and 18.6%, respectively, for CH_3_OO^•^ scavenging (Table 2b). Because the same type of bond, i.e., phenolic O–H bond, is involved in the underlying PCET mechanism, BDE correctly indicates the difference in reactivity. The path via the 4′-OH group is the one contributing the most to HOO^•^ scavenging (77.6%) and CH_3_OO^•^ scavenging (65.4%). As already emphasized, the most important parameter for estimating the scavenging potency of a phenolic compound is the rate constant for H-atom donation. Log *k*^TST^ and log *k*^TST/Eck^ are highly correlated with the BDE of rooperol’s O–H bonds: *r* = −0.9652 and −0.9570, respectively, for HOO^•^ scavenging, and *r* = −0.9695 and −0.9401, respectively, for CH_3_OO^•^ scavenging (Appendix A). Rate constants (*k*^TST/Eck^) of rooperol’s A-ring catecholic 3′-OH and 4′-OH groups are approximately one order of magnitude higher than the corresponding 3″-OH and 4″-OH groups related to the B-ring (Table 2). This could be ascribed to the influence of the proximate part of *para*-substituted chain (Figure 1). Electron-donating substituents reduce the BDE of the O–H bond, thus contributing to a faster H-atom donation [14]. The 1,2-Ethenediyl group (–C=C–) as an electron-donating group attached to the A-ring of rooperol affords an easier hydrogen atom abstraction from the catecholic OH groups than an electron-withdrawing ethyndiyl group (–C≡C–) attached to the B-ring. This contributes to reducing the Δ*G*^≠^ value for A-ring 3′-OH (4′-OH) groups by 1.3 (1.8) kcal/mol for HOO^•^ scavenging (and 0.6 (1.4) kcal/mol for CH_3_OO^•^ scavenging) in comparison with the B-ring catecholic moiety (Table 2).

The reaction pathways via vinylic hydrogens are predicted to be significantly endergonic (Δ_r_*G* > 17 kcal/mol) and consequently thermodynamically unfeasible. Therefore, these pathways can be ruled out as feasible for HOO^•^ scavenging by rooperol. As expected, performed kinetic analysis (Table 2) indicates that HAT from vinylic hydrogens is irrelevant, in accordance with the literature data [4].

In analyzing phenolic O–H vs. allylic C–H reactivity, consideration of the BDE is useless because the former has roughly 3 kcal/mol lower barrier heights, which results in ~10^3^ faster H-atom donation from phenolic O–H bonds (Table 2a). A similar tunnelling contribution (*κ*^Eck^ and *κ*^Wig^) nearly equally (~10^2^) enhances rate constants, keeping phenolic H-atom transfer much faster. Therefore, in the case of rooperol, Δ*G*^≠^ values make the difference in phenolic O–H vs. allylic C–H reactivity, while tunnelling additionally enhances the rate of reactions. Thus, regarding kinetics and its relation to BDE, it should be emphasized that a comparison of BDE values makes sense only if the same type of bond is considered, as emphasized in the case of phenolic O–H bonds of quercetin. As can be seen from Appendix A, taking BDE values of O–H and C–H bonds together is inappropriate in the estimation of kinetic information. No reasonable correlation was found between the reaction rates and the BDEs of different types of H-atom donors because O–H bond breaking and C–H bond breaking fall on different Evans–Polanyi correlation lines [17].

### 2.3. SOMO at TS of H-Atom Transfer from Phenolic O–H and Allylic C–H Groups

It has been suggested that an H-atom transfer that involves only one or no heteroatom must be HAT, while PCET requires an H-atom exchange between heteroatoms [10]. The studied mechanisms in previous sections were in accordance with this statement. To confirm the validity of such an approach, the singly occupied molecular orbital (SOMO) at the TS of H-atom transfer from the phenolic O–H and allylic C–H group was analyzed in this section. This is the simplest method to ascertain the underlying mechanism of H-atom transfer to radicals [36].

As can be seen from Figure 3a, where the transfer of H-atom from the 3-OH group of quercetin to HOO^•^ radical is analyzed, the SOMO at the TS involves 2p−π orbitals that are nearly orthogonal to the O-donor···H···O-acceptor axis, indicating the PCET mechanism as operative. A proton is transferred between lone pairs of electrons in σ orbitals on the oxygens, and an electron is synchronously transferred from the doubly occupied 2p−π orbital on the oxygen of quercetin’s 3-OH group to the singly occupied 2p−π orbital on the oxygen of HOO^•^ radical. The proton donor–acceptor distance at the TS is 2.36 Å, which enables a strong hydrogen bond along the O···H···O axis, similar to that of the phenol/phenoxyl reaction (2.40 Å), previously designated as a typical PCET [36].

A different picture arises at the SOMO of the TS for the allylic hydrogen/^•^OOH system (Figure 3b). Analysis of the shape of the SOMO density surfaces of the TS reveals that HAT is the prevailing reaction pathway since the SOMO is dominated by the atomic orbitals oriented along the C···H···^•^OOH transition vector. In this case, a hydrogen atom is transferred as a single particle from allylic moiety to the ^•^OOH. The donor–acceptor distance C···H···O at the TS (2.56 Å) resembles that of the toluene/benzyl reaction (2.72 Å), previously designated as HAT [36].

### 2.4. SET from Phenoxide Anions of Quercetin and Rooperol to HOO^•^ and CH_3_OO^•^ Radicals

In polar solutions, the SET mechanism is recognized as operative and much faster than HAT/PCET [37]. It proceeds via electron donation from the deprotonated phenolic OH group, i.e., phenoxide anion –O^−^. There is much discrepancy regarding the p*K*_a_s of quercetin as well as the order of OH group acidity [38,39]. Usually, with few exceptions, the 7-OH group is considered the most acidic, followed by the 4′-OH and 3-OH groups.

According to a study by Alvarez-Diduk et al., the order for the first three deprotonation steps of quercetin is 4′-OH, 7-OH and 3-OH, with the respective p*K*_a_ values of 6.41, 7.81 and 10.19 [39]. This implies that at the physiological pH of 7.4, the molar fractions of quercetin species amount as follows: AH = 0.0686, A^−^ = 0.6702, A^2−^ = 0.2608 and A^3−^ = 0.0004, Appendix A. The first deprotonation site of quercetin (4′-OH group) is expected to be the one contributing the most to the HOO^•^ and CH_3_OO^•^ scavenging via the SET mechanism.

Some of the calculated rate constants for the SET mechanism (*k*^TST^) were close to or even larger than the diffusion limit. Any rate constant larger than the diffusion rate (*k*_D_) lacks physical meaning. Because of that, the apparent rate constant (*k*_app_) for each reaction path was calculated and is presented in Table 3. The *k*_app_ more realistically reproduces the actual behavior under experimental conditions and enables comparison with experimentally measured rate constants [16]. Because kinetics is influenced by the abundance of the reactants, their molar fractions (*f*_M_) must be considered in order to calculate a rate constant that can be directly compared to the assayed one at the pH of interest [16]: *k*_Mf_ = *f*_A−_ × *f*_HOO•_ × *k*_app_. The rate constants *k*_Mf_ given in Table 3 include the molar fraction of HOO^•^ (*f*_HOO•_ = 0.00251) and quercetin’s monoanionic species (*f*_A−_ = 0.6702) at pH = 7.4. By taking this into account and by summing up the *f*_HOO•_ × *f*_A−_ × *k*_app_ for all reaction paths, the calculated overall *k*_Mf_ value for monoanions amounts to 5.8 × 10^5^ M^−1^ s^−1^. Consequently, quercetin monoanionic species (3-O^−^ > 3′-O^−^ ≈ 4′-O^−^) should be considered potent HOO^•^ scavengers via the SET mechanism at physiological pH in an aqueous environment.

Results presented in Table 3 indicate that the main contribution to HOO^•^ and CH_3_OO^•^ scavenging by quercetin’s monoanions is from 3-O^−^ phenoxide anion (Γ > 90%), followed by catecholic moiety (Γ < 10%). The highest reactivity of the 3-O^−^ anion could be related to the possibility that the deprotonation of the OH group from all positions can occur simultaneously, which allows the simultaneous presence of several monoanionic forms of quercetin [40]. A small fraction of the deprotonated 3-OH group, due to a larger driving force for electron transfer than from the more acidic 4′-OH group, is thermodynamically and kinetically important [38]. In this way, the 3-O^−^ phenoxide anion of quercetin reacts fastest with HOO^•^ and CH_3_OO^•^ in water as a solvent. Our calculated rate constant for HOO^•^ scavenging ($k_{\mathrm{Mf}}^{\mathrm{total}\;A-}$ = 5.8 × 10^5^ M^−1^ s^−1^) is in line with the experimentally determined quercetin’s reactivity with peroxyl radicals in water at pH = 7.4, *k* = 1.6 × 10^5^ M^−1^ s^−1^ [8]. Authors indicated that a rate-determining reaction with peroxyl radicals occurs from the equilibrating monoanions and involves firstly the 3-O^−^ phenoxide anion, as recognized by Musialik et al. [38]. This is in line with the observation that hydroxylation at the 3-position strongly increases the ability of a flavonoid to donate electrons [41]. In summary, the antiradical potency of quercetin in aqueous solution via the SET mechanism overwhelms its scavenging ability in non-polar media via the PCET mechanism.

By carefully checking the literature data, we were not able to find an experimental rate constant for HOO^•^ and CH_3_OO^•^ scavenging by quercetin in neutral water solution. For the purpose of comparison, i.e., to illustrate the ability of quercetin’s anions in the scavenging of HOO^•^, we used a published rate constant for the reaction of HOO^•^ with ascorbate ions in aqueous solution at pH = 7.31. The observed value was 3.1 × 10^5^ M^−1^ s^−1^ [42]. By considering the acid–base equilibrium of quercetin and ^•^OOH, i.e., the molar fractions of A^−^ and HOO^•^ at pH = 7.31 (*f*_A−_ = 0.6932, *f*_HOO•_ = 0.00308) and related *k*_app_ values, the rate constant for HOO^•^ inactivation amounts to *k*_Mf_ = 7.3 × 10^5^ M^−1^ s^−1^. Therefore, our kinetic data indicate quercetin as slightly more potent than ascorbic acid in the scavenging of HOO^•^ radicals, in accordance with the experimental results of the scavenging of ABTS radicals under the same conditions (VCEAC of 229.4 mg/L vs. 100 mg/L, respectively) [43].

The results of the performed kinetic analysis for the SET mechanism related to rooperol’s phenoxide anions are also given in Table 3. To the best of our knowledge, the experimental p*K*_a_ values of rooperol have not been reported yet. The prediction made by the ACD/pKa GALAS algorithm indicates the 4′-OH group as the most acidic (p*K*_a1_ = 9.7) followed by the 3″-OH group (p*K*_a2_ = 10.3) and 4″-OH group (p*K*_a3_ = 13.5). This implies that at the physiological pH of 7.4, the molar fractions of rooperol species amount as follows: AH = 0.9950, A^−^ = 5.00 × 10^−3^ and A^2−^ = 6.28 × 10^−6^.

The rooperol’s A-ring 4′-O^−^ phenoxide anion appears as the preferred site for HOO^•^ scavenging via the SET mechanism (Γ = 82.6%) (Table 3), analogously as in the case of HOO^•^ inactivation in non-polar media via the PCET mechanism (Γ^Eck^ = 77.6%) (Table 2). Similar results were obtained for the scavenging of CH_3_OO^•^. Consequently, the catecholic A-ring of rooperol is much more active than the catecholic B-ring in peroxyl quenching in both polar and non-polar media. It should be pointed out that monoanions of rooperol appear as nearly equally potent scavengers of HOO^•^ as well as monoanions of quercetin ($k_{\mathrm{overall}\;}^{\mathrm{Mf}}$ = 7.4 × 10^4^ vs. 5.8 × 10^5^ M^−1^ s^−1^, respectively). This result is in line with in vitro assayed activity of rooperol and quercetin in the scavenging of DPPH [44]. Additionally, the nearly equal activity of rooperol and catechins was experimentally determined in in vitro scavenging of ABTS radicals in a water environment [6].

It should be emphasized that the calculated rate constants *k*_Mf_ in aqueous media are higher for the reactions involving CH_3_OO^•^ compared with the corresponding rate constants involving HOO^•^. The acid–base equilibrium of HOO^•^ in aqueous solution (p*K*_a_ = 4.8) is largely responsible for this increase. Because kinetics is influenced by the abundance of the reactants, the molar fraction of HOO^•^ at the pH of interest (at pH = 7.4 it amounts 0.00251) must be included in the calculations, while it is ignored in the case of CH_3_OO^•^, which has no acid–base equilibria [45]. This is the main reason CH_3_OO^•^ reacts faster via the SET mechanism than the HOO^•^ in a water environment.

The experimentally measured rate constants of the reactions of HOO^•^ with PUFAs (linoleic, linolenic and arachidonic acid) are in the range 1.18–3.05 × 10^3^ M^−1^ s^−1^, assayed in aqueous ethanolic solutions at very low pH [4]. Compounds that react faster with HOO^•^ than the double allylic hydrogens of the PUFAs are expected to act as efficient antioxidants in suppressing peroxyl oxidation of membrane lipids as well as damage to the proteins and DNA because the reactivity of these biological targets is lower than that of PUFAs [16]. In polar media, data presented in Table 3 indicate quercetin and rooperol as efficient protectors of biological macromolecules from peroxyl cellular damage ($k_{\mathrm{overall}}^{\mathrm{Mf}}$ > 10^4^ M^−1^ s^−1^). In non-polar media their predicted protective ability is reduced ($k_{\mathrm{overall}}^{\mathrm{TST}/\mathrm{Eck}}$ ≤ 1.4 × 10^3^ M^−1^ s^−1^).

## 3. Materials and Methods

Geometry optimizations and frequency calculations for quercetin and rooperol and their species involved in the studied H-atom abstraction reactions in pentyl ethanoate and single electron donation reactions in water were carried out using the Gaussian 09 program package at the M05-2X/6-311++G(d,p) level of theory [46]. The best combination of the DFT functional/basis set/solvation model can be identified for each antioxidant by comparison with experimentally available data. To the best of our knowledge, kinetic data for CH_3_OO^•^ and HOO^•^ scavenging by studied molecules do not exist. Amongst density functionals designed for thermochemistry, reaction kinetics and noncovalent interactions involving molecules composed of main-group elements, we chose the M05-2X functional [47]. The M05-2X provides very good performance for thermochemistry and barrier heights [48]. Hence, it is particularly suitable to study the antioxidant properties and the reaction mechanisms involved in radical scavenging by polyphenols. The influence of pentyl ethanoate and water as solvents was studied using an implicit continuum solvation model—SMD [49]—which considers the full solute electron density in the estimation of solvation energy. SMD in conjunction with the M05-2X density functional was successfully used for study of the thermodynamics and kinetics of radical scavenging mechanisms [16]. The chosen level of theory, i.e., M05-2X/6-311++G(d,p)/SMD, was recommended by the designers of the QM-ORSA protocol, which has been designed and successfully used for the accurate prediction of rate constants for reactions of polyphenols with radicals [16]. Local minima and transition state (TS) were identified by the number of imaginary frequencies: local minima have only real frequencies, while TS was identified by the presence of a single imaginary frequency. Intrinsic reaction coordinate (IRC) calculation was performed on both sides of the TS to confirm that it properly connects two corresponding energy minima: reactant complex (RC) and product complex (PC). Further optimizations were carried out on the IRC final structures in order to obtain fully relaxed geometries. All computations were performed at 298.15 K. BDE related to the H-atom abstraction was calculated as described elsewhere [50]. p*K*_a_ values were predicted by using the ACD/pKa GALAS algorithm [51].

The rate constants (*k*^TST^) for fHAT reactions were calculated by using the conventional transition state theory (TST) as implemented in the Eyringpy program [26] according to Equation (3):(3)$$k^{\mathrm{TST}}=\sigma \kappa \frac{k_{B}T}{h}e^{-\left({\Delta G}^{\neq }\right)/RT}$$
σ is the reaction path degeneracy, i.e., the number of different but equivalent possible reaction pathways, κ accounts for tunnelling corrections, *k*_B_ is the Boltzmann constant, *T* is the thermodynamic temperature, *h* is the Planck constant and Δ*G*^≠^ is the Gibbs free activation energy of the studied reaction. Estimation of the Eckart tunnelling corrections may also take into consideration RC and PC [52].

For the SET reactions, the Marcus theory was used [53]. It relies on the transition state formalism and allows calculation of the barrier of any SET reaction from two thermodynamic parameters, the free energy of reaction, $\Delta G_{\mathrm{SET}}^{0}$, and the nuclear reorganization energy, λ: (4)$$\Delta G_{\mathrm{SET}}^{\neq }=\frac{\lambda }{4}{\left(1+\frac{\Delta G_{\mathrm{SET}}^{0}}{\lambda }\right)}^{2}$$
(5)$$\lambda \;\approx \;\Delta E_{\mathrm{SET}}-\Delta G_{\mathrm{SET}}^{0}$$
$\Delta E_{\mathrm{SET}}$ is the nonadiabatic energy difference between reactants and vertical products for SET. Accordingly, the TST rate constant for SET reactions is computed in the Eyringpy program using Equation (6):(6)$$k_{\mathrm{SET}}=\frac{k_{B}T}{h}e^{-\left(\Delta G_{\mathrm{SET}}^{\neq }\right)/RT}$$

Some of the rate constants calculated using the conventional TST can be sometimes equal to or even higher than the diffusion-limited rate constant. In this case, the kinetics of the reaction is controlled by the rate at which reactants diffuse towards each other. To preserve the physical meaning, the reaction rate constant must be smaller than the diffusion limit [16]. In this case, the apparent rate constant (*k*_app_), which is expected to reproduce the experimental findings, and rate constant for an irreversible bimolecular diffusion-controlled reaction (*k*_D_) were calculated.

The branching ratios (Γ), calculated from the rate constants, can be used to identify the reaction pathways contributing most to the total reaction (in %) [16]. They are calculated as:(7)$$\Gamma =100\;\frac{k_{i}}{k_{\mathrm{overall}}}$$
where *k_i_* represents the rate constant of an independent path. The overall rate constant (*k*_overall_) is calculated as the sum of rate constants of all reaction paths.

The distinction between the HAT and PCET mechanisms was analyzed by consideration of the character of the SOMO at the H-abstraction TS [16,36].

## 4. Conclusions

The impact of the electronic structure of quercetin and rooperol on the inactivation of CH_3_OO^•^ and HOO^•^ radicals via the fHAT and SET mechanisms was investigated in lipid and aqueous environments. Quercetin’s catecholic moiety plays a major role in the scavenging of both radicals via PCET, while its 3-O^−^ phenoxide anion is the most active via SET. Rooperol’s catecholic A-ring is the preferred site for the inactivation of CH_3_OO^•^ and HOO^•^ via PCET and SET pathways. Compared to quercetin, rooperol shows nearly equal scavenging potency in both media. Results of the performed kinetic analysis confirm the traditional view that phenolic OH groups play a central role in radical scavenging by polyphenols. Our results clearly indicate that the contribution of the aliphatic hydrogens of polyphenols to the antioxidant potency is negligible in accordance with known facts related to the O–H vs. C–H reactivity. We showed that predictions of reactivity based on the BDE values could be questionable. Undoubtedly, thermodynamically based predictions should be supported by kinetic analysis including the quantum mechanical tunnelling.

## Figures and Tables

**Figure 1 ijms-23-14497-f001:**
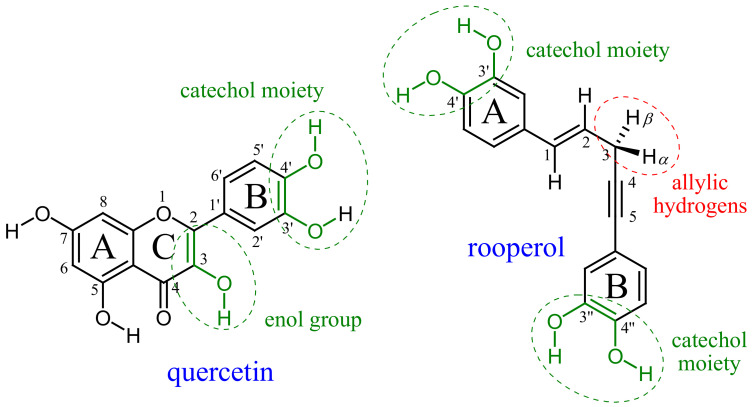
Structure, atom numbering and antiradical moieties (in green) of quercetin and rooperol.

**Figure 2 ijms-23-14497-f002:**
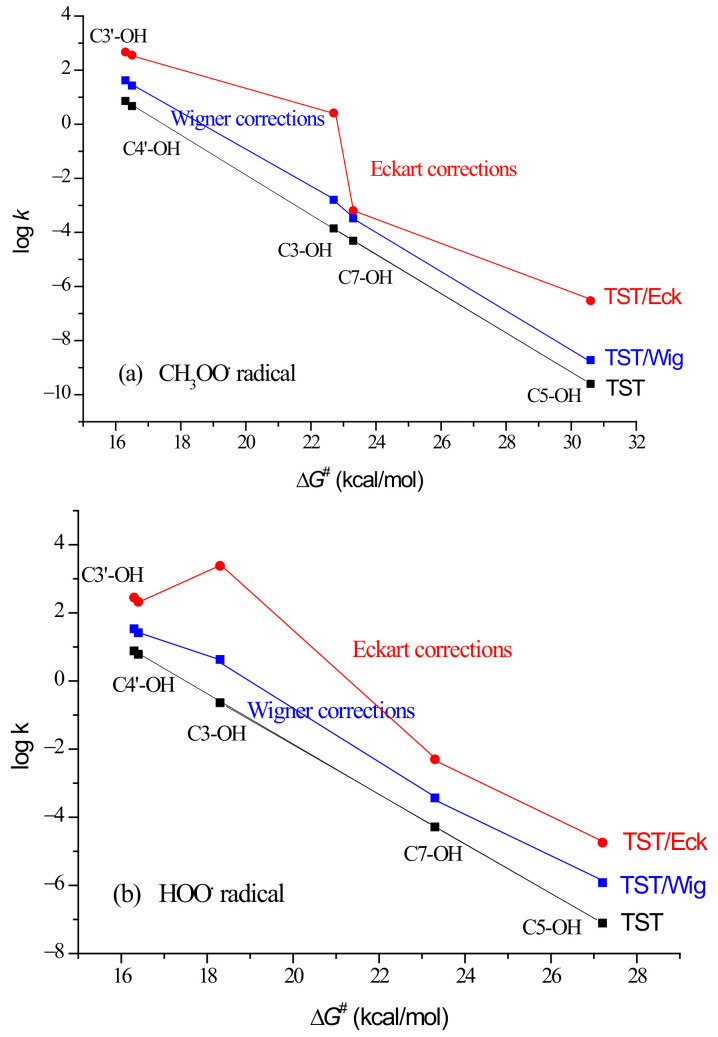
Influence of tunnelling corrections on the kinetics of the reaction between quercetin’s phenolic OH groups and: (**a**) CH_3_OO^•^; (**b**) HOO^•^.

**Figure 3 ijms-23-14497-f003:**
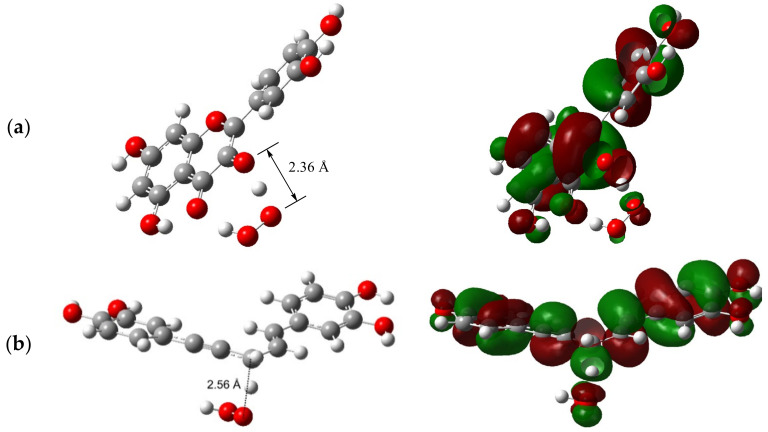
The TS structure accompanied by the corresponding SOMO for the reaction of: (**a**) 3-OH group of quercetin with HOO^•^; (**b**) allylic hydrogen of rooperol with HOO^•^.

**Table 1 ijms-23-14497-t001:** Bond dissociation enthalpy (BDE) (kcal/mol), reaction Gibbs free energy Δ_r_*G* (kcal/mol), TS imaginary frequency *ν* (cm^−1^), Gibbs free energy of activation Δ*G*^≠^ (kcal/mol), TST rate constant *k*^TST^ (M^−1^ s^−1^), Eckart (Wigner) tunnelling correction *κ*^Eck^ (*κ*^Wig^), Eckart (Wigner) rate constant *k*^TST/Eck^ (*k*^TST/Wig^) (M^−1^ s^−1^) and branching ratio Γ (%) for the PCET paths of quercetin with CH_3_OO^•^ and HOO^•^, in pentyl ethanoate at 298.15 K. $k_{\mathrm{overall}}^{\mathrm{TST}},\;k_{\mathrm{overall}}^{\mathrm{TST}/\mathrm{Eck}}$ and $k_{\mathrm{overall}}^{\mathrm{TST}/\mathrm{Wig}}$ are the sums of the rate constants of all reaction paths.

Path	BDE	Δ_r_G	ν	Δ*G*^≠^	*k* ^TST^	*κ* ^Eck^	*k* ^TST/Eck^	Γ^Eck^	*κ* ^Wig^	*k* ^TST/Wig^	Γ^Wig^
**CH_3_OO^•^**											
3-OH	85.41	0.9	−3307	22.7	1.4 × 10^−4^	18350.5	2.6 × 10^0^	0.3	11.6	1.6 × 10^−3^	0
5-OH	99.41	14.9	−2631	30.6	2.5 × 10^−10^	120.0	3.0 × 10^−8^	0	7.7	1.9 × 10^−9^	0
7-OH	94.55	10.4	−2456	23.3	4.9 × 10^−5^	13.2	6.4 × 10^−4^	0	6.9	3.3 × 10^−4^	0
3′-OH	82.36	−1.1	−2227	16.3	7.2 × 10^0^	65.8	4.7 × 10^2^	56.4	5.8	4.2 × 10^1^	60.0
4′-OH	80.35	−3.5	−2259	16.5	4.7 × 10^0^	77.1	3.6 × 10^2^	43.2	6.0	2.8 × 10^1^	40.0
			$k_{\mathrm{overall}}^{\mathrm{TST}}$ = 1.2 × 10^1^			$k_{\mathrm{overall}}^{\mathrm{TST}/\mathrm{Eck}}$ = 8.3 × 10^2^			$k_{\mathrm{overall}}^{\mathrm{TST}/\mathrm{Wig}}$ = 7.0 × 10^1^		
**HOO^•^**											
3-OH	85.41	−1.3	−4274	18.3	2.3 × 10^−1^	10301.8	2.4 × 10^3^	83.0	18.7	4.3 × 10^0^	6.7
5-OH	99.41	12.7	−3894	27.2	7.8 × 10^−8^	233.0	1.8 × 10^−5^	0	15.7	1.2 × 10^−6^	0
7-OH	94.55	8.1	−2521	23.3	5.2 × 10^−5^	96.8	5.0 × 10^−3^	0	7.2	3.7 × 10^−4^	0
3′-OH	82.36	−3.4	−1884	16.3	7.6 × 10^0^	36.7	2.8 × 10^2^	9.7	4.4	3.4 × 10^1^	52.9
4′-OH	80.35	−5.8	−1843	16.4	6.1 × 10^0^	34.0	2.1 × 10^2^	7.3	4.3	2.6 × 10^1^	40.4
			$k_{\mathrm{overall}}^{\mathrm{TST}}$ = 1.4 × 10^1^			$k_{\mathrm{overall}}^{\mathrm{TST}/\mathrm{Eck}}$ = 2.9 × 10^3^			$k_{\mathrm{overall}}^{\mathrm{TST}/\mathrm{Wig}}$ = 6.4 × 10^1^		

**Table 2 ijms-23-14497-t002:** Bond dissociation enthalpy BDE (kcal/mol), reaction Gibbs free energy Δ_r_*G* (kcal/mol), TS imaginary frequency *ν* (cm^−1^), Gibbs free energy of activation Δ*G*^≠^ (kcal/mol), TST rate constant *k*^TST^ (M^−1^ s^−1^), Eckart (Wigner) tunnelling correction *κ*^Eck^ (*κ*^Wig^), Eckart (Wigner) rate constant *k*^TST/Eck^ (*k*^TST/Wig^) (M^−1^ s^−1^) and branching ratio Γ (%) for the PCET paths of rooperol with HOO^•^ and CH_3_OO^•^, in pentyl ethanoate at 298.15 K. $k_{\mathrm{overall}}^{\mathrm{TST}},\;k_{\mathrm{overall}}^{\mathrm{TST}/\mathrm{Eck}}$ and $k_{\mathrm{overall}}^{\mathrm{TST}/\mathrm{Wig}}$ are the sums of the rate constants of all reaction paths. For denotation of allylic and vinylic reaction paths see Figure 1.

Path	BDE	Δ_r_G	ν	Δ*G*^≠^	*k* ^TST^	*κ* ^Eck^	*k* ^TST/Eck^	Γ^Eck^	*κ* ^Wig^	*k* ^TST/Wig^	Γ^Wig^
**(a) HOO^•^**											
**phenolic reaction paths**											
3′-OH	80.48	−4.8	−1801	16.3	7.4 × 10^0^	23.5	1.7 × 10^2^	12.0	4.1	3.1 × 10^1^	7.8
4′-OH	77.04	−8.0	−1616	14.7	9.7 × 10^1^	11.1	1.1 × 10^3^	77.6	3.5	3.4 × 10^2^	86.1
3″-OH	81.81	−3.0	−1817	17.6	8.2 × 10^−1^	30.6	2.5 × 10^1^	1.7	4.2	3.5 × 10^0^	0.9
4″-OH	79.63	−5.3	−1761	16.5	5.0 × 10^0^	24.6	1.2 × 10^2^	8.5	4.0	2.0 × 10^1^	5.1
			$k_{\mathrm{overall}}^{\mathrm{TST}}$ = 1.1 × 10^2^			$k_{\mathrm{overall}}^{\mathrm{TST}/\mathrm{Eck}}$ = 1.4 × 10^3^		99.8	$k_{\mathrm{overall}}^{\mathrm{TST}/\mathrm{Wig}}$ = 3.9 × 10^2^		99.9
**allylic reaction paths**											
allylic H*α*	72.21	−14.1	−1739	19.4	3.6 × 10^−2^	39.6	1.4 × 10^0^	0.1	3.9	1.4 × 10^−1^	0.03
allylic H*β*	72.21	−14.1	−1730	19.3	4.3 × 10^−2^	37.0	1.6 × 10^0^	0.1	3.9	1.7 × 10^−1^	0.04
			$k_{\mathrm{overall}}^{\mathrm{TST}}$ = 7.9 × 10^−2^			$k_{\mathrm{overall}}^{\mathrm{TST}/\mathrm{Eck}}$ = 3.0 × 10^0^		0.2	$k_{\mathrm{overall}}^{\mathrm{TST}/\mathrm{Wig}}$ = 3.1 × 10^−1^		0.07
**vinylic reaction paths**											
vinylic H1	101.07	17.7	−1881	32.8	5.9 × 10^−12^	36.8	2.2 × 10^−10^	0.0	4.4	2.6 × 10^−11^	0.00
vinylic H2	109.70	23.8	−1913	33.5	1.6 × 10^−12^	0.8	1.3 × 10^−12^	0.0	4.6	7.3 × 10^−12^	0.00
			$k_{\mathrm{overall}}^{\mathrm{TST}}$ = 7.5 × 10^−12^			$k_{\mathrm{overall}}^{\mathrm{TST}/\mathrm{Eck}}$ = 2.2 × 10^−10^		0.0	$k_{\mathrm{overall}}^{\mathrm{TST}/\mathrm{Wig}}$ = 3.3 × 10^−11^		0.00
**(b) CH_3_OO^•^**											
3’-OH	80.48	−2.6	−2085	17.1	1.7 × 10^0^	45.0	7.8 × 10^1^	16.0	5.2	9.1 × 10^0^	9.5
4’-OH	77.04	−5.7	−1866	15.8	1.7 × 10^1^	18.4	3.2 × 10^2^	65.4	4.4	7.6 × 10^1^	79.3
3’’-OH	81.81	−0.7	−2087	17.7	6.7 × 10^−1^	53.6	3.6 × 10^1^	7.4	5.2	3.5 × 10^0^	3.7
4’’-OH	79.63	−3.0	−1992	17.2	1.5 × 10^0^	36.9	5.5 × 10^1^	11.2	4.8	7.2 × 10^0^	7.5
			$\;k_{\mathrm{overall}}^{\mathrm{TST}}$ = 2.0 × 10^1^			$k_{\mathrm{overall}}^{\mathrm{TST}/\mathrm{Eck}}$ = 4.9 × 10^2^		100.0	$k_{\mathrm{overall}}^{\mathrm{TST}/\mathrm{Wig}}$ = 9.6 × 10^1^		100.0

**Table 3 ijms-23-14497-t003:** Reaction Gibbs free energy (Δ_r_*G*, kcal/mol), Gibbs free energy of activation (Δ*G*^≠^, kcal/mol), reorganization energy (λ, kcal/mol), TST rate constant (*k*^TST^, M^−1^ s^−1^) diffusion rate constant (*k*_D_, M^−1^ s^−1^), apparent rate constant (*k*_app_, M^−1^ s^−1^), rate constant including molar fractions of radical and phenoxide anion (*k*_Mf_, in M^−1^ s^−1^), and branching ratio (Γ, %) for the SET reactions between phenoxide anions of: (a) quercetin, and (b) rooperol with ^•^OOH and CH_3_OO^•^, in water at pH = 7.40 and 298.15 K.

Path	Δ_r_*G*	Δ*G*^≠^	λ	*k* ^TST^	*k* _D_	*k* _app_	*k* _Mf_	Γ
**(a) scavenging of ^•^OOH by quercetin phenoxide anions**								
3-O^−^	3.0	5.9	16.9	3.2 × 10^8^	8.2 × 10^9^	3.1 × 10^8^	5.2 × 10^5^	90.3
5-O^−^	13.4	13.5	15.1	8.0 × 10^2^	8.3 × 10^9^	8.0 × 10^2^	1.3 × 10^0^	0
7-O^−^	18.5	18.8	14.3	1.0 × 10^−1^	8.2 × 10^9^	1.0 × 10^−1^	1.7 × 10^−4^	0
3′-O^−^	5.8	7.5	16.2	2.0 × 10^7^	8.2 × 10^9^	2.0 × 10^7^	3.4 × 10^4^	5.7
4′-O^−^	6.3	7.7	15.6	1.4 × 10^7^	8.2 × 10^9^	1.4 × 10^7^	2.4 × 10^4^	4.0
						$k_{\mathrm{overall}}^{\mathrm{Mf}}$ =	5.8 × 10^5^	
**scavenging of CH_3_OO^•^ by quercetin phenoxide anions**								
3-O^−^	4.9	6.9	16.5	5.1 × 10^7^	7.7 × 10^9^	5.1 × 10^7^	3.4 × 10^7^	99.8
5-O^−^	15.4	15.4	14.7	3.4 × 10^1^	7.8 × 10^9^	3.4 × 10^1^	2.3 × 10^1^	0
7-O^−^	20.4	21.2	13.9	1.9 × 10^−3^	7.8 × 10^9^	1.9 × 10^−3^	1.3 × 10^−3^	0
3′-O^−^	7.8	8.8	15.8	2.3 × 10^6^	7.7 × 10^9^	2.3 × 10^6^	1.5 × 10^6^	0.1
4′-O^−^	8.3	9.0	15.2	1.5 × 10^6^	7.7 × 10^9^	1.5 × 10^6^	1.0 × 10^6^	0.1
						$k_{\mathrm{overall}}^{\mathrm{Mf}}$ =	3.7 × 10^7^	
**(b) scavenging of ^•^OOH by rooperol phenoxide anions**								
3′-O^−^	2.5	5.4	16.4	6.3 × 10^8^	8.1 × 10^9^	5.8 × 10^8^	7.3 × 10^3^	9.8
4′-O^−^	−0.8	3.7	16.3	1.2 × 10^10^	8.2 × 10^9^	4.9 × 10^9^	6.1 × 10^4^	82.6
3″-O^−^	3.5	6.3	17.5	1.6 × 10^8^	8.3 × 10^9^	1.5 × 10^8^	1.9 × 10^3^	2.5
4″-O^−^	3.4	5.9	15.9	3.1 × 10^8^	8.3 × 10^9^	3.0 × 10^8^	3.8 × 10^3^	5.1
						$k_{\mathrm{overall}}^{\mathrm{Mf}}$ =	7.4 × 10^4^	
**scavenging of CH_3_OO^•^ by rooperol phenoxide anions**								
3′-O^−^	4.5	6.5	16.0	1.0 × 10^8^	7.7 × 10^9^	1.0 × 10^8^	5.0 × 10^5^	4.4
4′-O^−^	1.1	4.6	15.8	2.8 × 10^9^	7.8 × 10^9^	2.1 × 10^9^	1.1 × 10^7^	92.5
3″-O^−^	5.4	7.4	17.0	2.4 × 10^7^	7.8 × 10^9^	2.4 × 10^7^	1.2 × 10^5^	1.0
4″-O^−^	5.3	7.0	15.5	4.7 × 10^7^	7.8 × 10^9^	4.7 × 10^7^	2.4 × 10^5^	2.1
						$k_{\mathrm{overall}}^{\mathrm{Mf}}$ =	1.1 × 10^7^	

## Data Availability

Data are contained within the article and Appendix A.

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
