# Peer review of "DFT Study of the Direct Radical Scavenging Potency of Two Natural Catecholic Compounds"

_ijms, 2022, doi:10.3390/ijms232214497_

Round 1

Reviewer 1 Report

This is a very good piece of scientific work. The conducted computations are state-of-the-art, the results are well presented and the final conclusions are valid.

Probably the best recommendation for publishing this manuscript is that reading it increased my knowledge and will help improve my future research.

Two minor points:

On page 4: "Recently, experimental ESR measurements indicated that the unpaired electron of quercetin radical is mostly delocalized in the B-ring and partly on the AC rings [22]." Do your calculations are in line with this experimental finding?  Please provide a short comment.

Figure 3: The resolution is poor. Perhaps it got lowered after the submission when a PDF file for review was compiled by the system. If this is not the case, please provide an image in higher resolution.

Author Response

Comment 1.

On page 4: "Recently, experimental ESR measurements indicated that the unpaired electron of quercetin radical is mostly delocalized in the B-ring and partly on the AC rings [22]." Do your calculations are in line with this experimental finding?  Please provide a short comment.

Our answer:

Regarding quenching of HOOradical, by taking into account correction of overestimated Eckart tunneling for the H-atom abstraction from 3-OH group to HOO radical (Table 1 and Figure 2b) which results in more reliable kTST/Eck value for C3-OH of 1.2 × 101 M−1 s−1 (instead of 2.4 × 103 M−1 s−1), it follows that our calculations are in line with this experimental finding.

Namely, the sum of kTST/Eck values for C3’-OH and C4’-OH (2.8 × 102 M−1 s−1 and 2.1 ´ 102 M−1 s−1, respectively) amounts to 4.9 × 102 M−1 s−1 which indicates that branching ratio GEck for B-ring amounts to 97.6 % and for AC rings 2.4% (GWig for B-ring amounts to 93.3 % and for AC rings 6.7%).

The branching ratio G is proportional to abundance of quercetin radical delocalization.

Regarding scavenging of CH3OOradical, agreement with experimental finding also exists (Table 1): branching ratio GEck for B-ring amounts to 99.6 % and for AC rings 0.3% (GWig for B-ring amounts to 100.0 % and for AC rings 0.0%).

Comment 2.

Figure 3: The resolution is poor. Perhaps it got lowered after the submission when a PDF file for review was compiled by the system. If this is not the case, please provide an image in higher resolution.

Our answer:

Resolution of Figure 3 is in accordance with required. So, poor resolution is probably caused by system conversion to PDF. Our conversion of .docx file to PDF results in nice resolution of all figures.

Reviewer 2 Report

In this study, the authors explored the impact of the electronic structure of quercetin and rooperol on the inactivation of CH3OO• and HOO• radicals via fHAT and SET mechanisms in lipid and aqueous environments. The study suggests that Quercetin’s catecholic moiety plays a major role in the scavenging of both radicals via PCET, while its 3-O− phenoxide anion is the most active via SET.

Overall, it is a good study, and it will be worth being published after some revisions are pointed out as follows.

1.     While I see the potential importance of this study, it is also important to articulate the importance of the study to the readers why they have used M05-2X. Because, in computational chemistry, choice of method is very crucial, and in particular when using DFT, choice of functional is very critical. So, using a particular functional for exploring a chemical reaction is not a good idea (unless there is a benchmark study on the similar/same system. In addition, it has been reported earlier, that several DFT-D functionals provide more accurate interaction energies than M05-2X. In Table 2. Authors Bond dissociation enthalpy BDE (kcal/mol), reaction Gibbs free energy DrG (kcal/mol), etc. These were computed with M05-2X. I would suggest author could check if using DFT-D functional can improve (or differ) the findings of this study. 

Author Response

Comment 1. a

While I see the potential importance of this study, it is also important to articulate the importance of the study to the readers why they have used M05-2X. Because, in computational chemistry, choice of method is very crucial, and in particular when using DFT, choice of functional is very critical. So, using a particular functional for exploring a chemical reaction is not a good idea (unless there is a benchmark study on the similar/same system.

Our answer:

The first paragraph of 4. Materials and Methods section is corrected by text indicated in blue colour:

Geometry optimizations and frequency calculations for quercetin and rooperol, and their species involved in the studied H-atom abstraction reactions in pentyl ethanoate and single electron donation reactions in water were carried out using the Gaussian 09 program package at the M05-2X/6-311++G(d,p)/SMD level of theory [46]. The best combination of the DFT functional/basis set/solvation model can be identified for each antioxidant by comparing with experimentally available data. To the best of our knowledge kinetic data for CH3OO and HOO scavenging by studied molecules does not exist. Amongst density functionals designed for thermochemistry, reaction kinetics and noncovalent interactions involving molecules composed of main-group elements, we chose the M05-2X functional [47]. The M05-2X provides very good performance for thermochemistry and barrier heights [48]. Hence, it is particularly suitable to study the antioxidant properties and the reaction mechanisms involved in the radical scavenging by polyphenols. The influence of pentyl ethanoate and water as solvents was studied using an implicit continuum solvation model – SMD [49], which considers the full solute electron density in the estimation of solvation energy. SMD in conjunction with the M05-2X density functional has been successfully used for study of thermodynamics and kinetics of radical scavenging mechanisms [16]. Chosen level of theory, i.e., M05-2X/6-311++G(d,p)/SMD, has been recommended by designers of the QM-ORSA protocol which has been designed and is currently successfully used for accurate prediction of rate constants for reactions of polyphenols with radicals [16]. …

Comment 1. b

In addition, it has been reported earlier, that several DFT-D functionals provide more accurate interaction energies than M05-2X. In Table 2. Authors Bond dissociation enthalpy BDE (kcal/mol), reaction Gibbs free energy DrG (kcal/mol), etc. These were computed with M05-2X. I would suggest author could check if using DFT-D functional can improve (or differ) the findings of this study.

Our answer:

It is a good idea to compare ability of M05-2X vs DFT-D functionals to reproduce some available assayed kinetic data for particular polyphenol(s). Therefore, it should be the subject of laborious work which may deserve publication in a new forthcoming publication.